# COVID-19 Vaccine Misinformation in Middle Income Countries

**Jongin Kim**[1], **Byeo Rhee Bak**[2], **Aditya Agrawal**[1], **Jiaxi Wu**[3],
**Veronika J. Wirtz**[4,5], **Traci Hong**[3,5], **Derry Wijaya**[1,6]

[1]Dept. of Computer Science, Boston University   [2]Dept. of Economics, Boston University
[3]College of Communication, Boston University   [4]School of Public Health, Boston University
[5]Center for Emerging Infectious Diseases Policy & Research, Boston University
[6]Monash University Indonesia
{jongin, brbak, adityaai, jiaxiw, vwirtz, tjhong, wijaya}@bu.edu

## Abstract

This paper introduces a multilingual dataset of COVID-19 vaccine misinformation, consisting of annotated tweets from three middle-income countries: Brazil, Indonesia, and Nigeria. The expertly curated dataset includes annotations for 5,952 tweets, assessing their relevance to COVID-19 vaccines, presence of misinformation, and the themes of the misinformation. To address challenges posed by domain specificity, the low-resource setting, and data imbalance, we adopt two approaches for developing COVID-19 vaccine misinformation detection models: domain-specific pre-training and text augmentation using a large language model. Our best misinformation detection models demonstrate improvements ranging from 2.7 to 15.9 percentage points in macro F1-score compared to the baseline models. Additionally, we apply our misinformation detection models in a large-scale study of 19 million unlabeled tweets from the three countries between 2020 and 2022, showcasing the practical application of our dataset and models for detecting and analyzing vaccine misinformation in multiple countries and languages. Our analysis indicates that percentage changes in the number of new COVID-19 cases are positively associated with COVID-19 vaccine misinformation rates in a staggered manner for Brazil and Indonesia, and there are significant positive associations between the misinformation rates across the three countries.

## 1 Introduction

Affluent countries have enjoyed ample supply of COVID-19 vaccines but distribution to low- and middle-income countries were slow. The void created by a slow vaccine roll out may create a space for misinformation to further flourish. Misinformation on social media is already documented to contribute to vaccine hesitancy in developed countries (Puri et al., 2020), but comparatively few studies have focused beyond high income settings (Hagg et al., 2018). Given that middle income countries (MIC) make up 75% of the world's population and 62% of the world's poor (World Bank, 2022), studying vaccine misinformation in MIC is important to preparing for future pandemics especially because global uptake to vaccines is needed to end pandemics (Asundi et al., 2021). Importantly, the impact of misinformation on vaccine hesitancy extends beyond COVID-19 in that there has been a precipitous decline in childhood vaccination rates in MIC (Guglielmi, 2022). The residual effects of vaccine misinformation on vaccine hesitancy presents a global health problem for containing infectious diseases.

In this study, we focus on three middle-income countries: Brazil, Indonesia, and Nigeria, which have the majority of their population using social media (Schumacher and Kent, 2020) and where vaccine confidence has eroded in recent years (De Figueiredo et al., 2020). In addition, each of these three countries are population hubs from three distinct regions in the world. While the majority of the population in these countries use social media (Guglielmi, 2022), social media usage is still projected to grow in these MIC countries whereas usage has plateaued in high-income countries (Poushter et al., 2018). This presents a unique opportunity to study the impact of social media on vaccine misinformation in MIC.

The development of an automated system that can detect vaccine misinformation will allow for epidemic surveillance in three heavily populated countries in distinct parts of the world. This will allow government agencies, health agencies, and researchers to monitor misinformation and coordinate intervention strategies to deter emerging infectious diseases. In addition, the automatic system for misinformation detection allows for identification of shared misinformation themes across three distinct countries/regions, which would enable health organizations to tailor health messages.

In this work, we make two main contributions. Firstly, we have curated a new multilingual dataset of COVID-19 vaccine misinformation from MIC, where social media misinformation is understudied. This dataset[1] is the first of its kind in several aspects: it is geolocated, in multiple languages, carefully curated, covers a wide time range from 2020 to 2022, and includes not only domain-expert annotations of misinformation, but also the identified themes based on a previous study of globally circulating COVID-19 vaccine information on online platforms (Islam et al., 2021). Secondly, utilizing this dataset, we have trained models that effectively detect misinformation and its associated themes, outperforming competitive baselines, including GPT-3 models. We apply our best models in a large scale study to analyze the trend of misinformation in the three MIC between 2020 and 2022. Our analysis yields interesting findings regarding the relationship between COVID-19 cases, vaccine misinformation rates, and variations across the three countries.

## 2 Related Works

While there have been studies on vaccine misinformation, no prior study has examined misinformation across countries or studied shared misinformation themes across MIC. Studies on vaccine misinformation utilizing Twitter data tend to be country-specific, predominantly focused on the United States (Pierri et al., 2022), restricted to the English language only (Featherstone et al., 2020), or lack specified geolocations (Argyris et al., 2022). Moreover, these studies are conducted during limited periods in 2021 (Pierri et al., 2022; Argyris et al., 2022), or in 2020 (Featherstone et al., 2020). Similarly, most previous works introducing new datasets related to COVID-19 misinformation primarily focus on English content from specific target locations (Hayawi et al., 2022; Muric et al., 2021; DeVerna et al., 2021; Weinzierl and Harabagiu, 2022b, 2021, 2022a; Hong et al., 2023), while Mubarak et al. (2022) released an Arabic tweet dataset covering countries in the Arab region.

Studies on vaccine misinformation are important as misinformation exposure contributes to increased vaccine hesitancy and reduced behavioral intention to get vaccinated (Lee et al., 2022; Dubé

et al., 2013). Media plays a prominent role in spreading misinformation. For instance, a longitudinal study on mother's attitudes towards MMR vaccines reveals a link between media-published misinformation, their perception of vaccine safety, and vaccination rates among children in the United Kingdom (Smith et al., 2007). Similarly, negative press in local media in the UK was associated with a decline in MMR vaccinations (Mason and Donnelly, 2000). The impact of misinformation on vaccination coverage is not limited to Europe; it is observed globally. A well-documented case is polio in Central and West Africa, where in the late 1990s and early 2000s misinformation about the polio vaccine spread within Muslim communities in Northern Nigeria, Benin, Burkina Faso, Cameroon, Central African Republic, and other neighboring countries (Jegede, 2007), leading to leaders refusing vaccination and subsequent outbreaks of the disease (WHO, 2006).

## 3 Dataset

We collect all publicly available Twitter posts in Brazil, Indonesia, and Nigeria that contain vaccine-related terms from January 2020 to December 2022 (N = 18,809,231). We use the hydration process (Arafat et al., 2021) where we use Brandwatch API to scrape IDs of tweets that contained predefined vaccine-related terms, then apply Twitter API to retrieve tweet contents.

Brandwatch enables us to scrape tweets specific to geolocations associated with the three focal countries. There are no language exclusions. In addition, we include "shot" as a search term because it is often used to describe vaccine, but we set limits to reduce data noise, particularly from references to parties and alcohol that are synonymous with the term "shot". Our complete list of search query is listed in Appendix A.1. In addition, we include country-specific terms including "vacina," "injecao," and "Zé Gotinha" for Brazil; and "vaksin" and "suntik" for Indonesia per the Oxford Languages Word for Vax (Oxford Languages, 2021).

For data annotation, we employ Quantitative Content Analysis (QCA) in communication research (Krippendorff, 2018) where a representative sample of the data is drawn and on which two or more trained coders (i.e., annotators) apply a codebook protocol that contains all the variables for annotation and their definitions. Similar to (Liu et al., 2019; Guo et al., 2021), prior to coding the

---

[1]The dataset and annotation codebook, which contains the operational definitions and examples of annotation variables including the misinformation themes, are available at https://github.com/zzoliman/covid-vaccine-misinfo-MIC

| Q | yes (%) | no (%) | uncertain (%) | total |
|---|---|---|---|---|
| Q1 | 3,666 (61.6) | 2,286 (38.4) | - | 5,952 |
| Q2 | 655 (17.9) | 3,011 (82.1) | - | 3,666 |
| Q3 | 1,119 (37.2) | 1,596 (53.0) | 296 (9.8) | 3,011 |
| Q4a | 186 (16.6) | 933 (83.4) | - | 1,119 |
| Q4b | 539 (48.2) | 580 (51.8) | - | 1,119 |
| Q4c | 26 (2.3) | 1,093 (97.7) | - | 1,119 |
| Q4d | 412 (36.8) | 707 (63.2) | - | 1,119 |
| Q4e | 106 (9.5) | 1,013 (90.5) | - | 1,119 |
| Q4f | 93 (8.3) | 1,026 (91.7) | - | 1,119 |
| Q4g | 272 (24.3) | 847 (75.7) | - | 1,119 |
| Q4h | 29 (2.6) | 1,090 (97.4) | - | 1,119 |

Table 1: The statistics of annotated tweets for each question. Q1: relevance to vaccine; Q2: mentions of specific non-COVID vaccine; Q3: presence of misinformation; and misinformation themes: Q4a: vaccine development, availability, or access; Q4b: safety, efficacy, or acceptance; Q4c: infertility; Q4d: political/economic motives; Q4e: mandatory vaccine and ethics; Q4f: vaccine reagents; Q4g: vaccine morbidity or mortality; and Q4h: vaccine alternatives.

entire sample independently, coders are trained on the codebook and their agreement on how to apply the codes is measured with inter-coder-reliability (ICR), where high values imply that the coders consistently annotate the data and signal the high validity of the annotations. Once they reach an acceptable ICR, coders code the rest of the sample independently. We use GWET's ICR measures (Gwet, 2008) where agreements are considered substantial (between 0.61–0.8) or near-perfect (between 0.81–0.99) (Landis and Koch, 1977).

To apply QCA, we randomly sample 5,500 tweets, stratified by the distribution of tweets collected in each of the three focal countries. Two communication graduate students are trained as coders to examine the tweets for: (Q1) relevance to vaccine (yes/no), (Q2) mentions of specific non-COVID vaccine (such as the MMR vaccine) (yes/no), (Q3) presence of misinformation (yes/no/uncertain). Content deemed to be misinformation is labeled further for its themes based on a pre-identified list of COVID-19 vaccine misinformation themes (Islam et al., 2021), which included misinformation pertinent to (Q4a) vaccine development, availability or access; (Q4b) safety, efficacy, or acceptance; (Q4c) infertility; (Q4d) political/economic motives; (Q4e) mandatory vaccine and ethics; (Q4f) vaccine reagents; (Q4g) vaccine morbidity or mortality; (Q4h) vaccine alternatives.

To assess inter-coder reliability, 550 posts are randomly selected from the data (N=5,500) and both trained coders independently code the selected posts. The mean reliability for the variables in our study is .88, with the following scores: (Q1) rel-

evance (0.98); (Q2) non-COVID vaccine (0.97); (Q3) misinformation (0.72). For misinformation themes: (Q4a) vaccine development, availability, or access (0.88); (Q4b) safety, efficacy or acceptance (0.88); (Q4c) infertility (0.89); (Q4d) political/economic motives (0.87); (Q4e) mandatory vaccine and ethics (0.88); (Q4f) vaccine reagents (0.87); (Q4g) vaccine morbidity or mortality (0.88); and (Q4h) vaccine alternatives (0.89). Upon determination that human coding is reliable, the trained coders independently code the remaining posts. Finally, tweets annotated with "uncertain" for misinformation are reviewed by our expert public health panel who make the final determination.

To balance the data for misinformation, we identify more tweets that contain misinformation, resulting in the final annotated dataset of N=5,952. Table 1 shows the statistics of the annotated dataset.

## 4 COVID-19 Vaccine Misinformation Models

In this section, we describe how we develop our COVID-19 vaccine misinformation models and present the experiment results.

### 4.1 Classification Models

To conduct a study on the large-scale tweets we collected using our vaccine-related query terms (§3), we first train a *vaccine relevance* model to distinguish tweets that are truly relevant to vaccines from the dataset. This model is trained using the annotations for Q1 (Table 1). Next, we train a *COVID-19 vaccine relevance* model to identify among vaccine-relevant tweets, tweets that are specific to COVID-19 vaccines. This model is trained using the annotations for Q2[2]. Then, we train a model to detect tweets containing *COVID-19 vaccine misinformation*. Training for this model was based on annotations for Q3. Tweets that were annotated to contain misinformation (Q3) were further annotated for whether they are related to a specific theme (Q4a to Q4h). We develop 8 binary classification models to identify *themes of misinformation* using the corresponding annotations.

### 4.2 Tweets Preprocessing

The labeled dataset was preprocessed before being used to train the models. All mentions of username

---

[2]The assumption made was that tweets that *do not mention* a specific non-COVID vaccine are COVID-19 vaccine-related tweets. This assumption takes into account that the tweets were collected during the COVID-19 pandemic.

were replaced with "@user" and all URLs were replaced with "http".

### 4.3 Pre-trained Model and Domain-specific Pre-training

In our experiment, we use XLM-RoBERTa (Conneau et al., 2020, XLM-R), which is a Transformer-based multilingual language model pre-trained on large multilingual corpus, as our base encoder for all our classification models. This model is chosen since our dataset contains tweets written from various languages including English, Portuguese, and Indonesian. English is the official and most widely spoken language in Nigeria and is also widely used in Brazil and Indonesia. Portuguese and Indonesian are the official and national language of Brazil and Indonesia, respectively.

Given that our experiment focuses on domain-specific data, specifically texts from tweets that are related to COVID-19 vaccine, we recognize the importance of using a pre-trained model optimized for this type of data. Therefore, we further pre-train the XLM-R[3] on the entirety of tweets we collect using our vaccine-related query terms and use it as another base encoder for our experiments. We refer to classification models that are fine-tuned on this encoder as XLM-R+. By pre-training the model on the domain-specific dataset, we aim to improve its performance and ability to accurately classify COVID-19 vaccine-related tweets. This approach has been shown to be effective in improving the performance of natural language processing models on domain-specific texts (Gururangan et al., 2020).

### 4.4 Text Augmentation

To alleviate the issue of imbalanced dataset when training models for classifying misinformation (Q3) and themes (Q4a to Q4h), we leverage GPT-3[4] (Brown et al., 2020), a large language model, to augment the training data for each model. In particular, given the limited availability of labeled positive examples, i.e., tweets containing misinformation and tweets associated with specific themes of misinformation (Table 1), we employ an augmentation technique to amplify the positive examples. Drawing inspiration from the work of Sahu et al. (2022), we employ a straightforward prompt-based approach to generate additional data. As illustrated

in Figure 1, we prompt GPT-3 with a set of 10 example tweets belonging to the same category and append a new example line for GPT-3 to generate tweet that is aligned with the target category and the examples provided. For each generation, the example tweets are randomly selected from positive examples in the training set. We refer to the model that is based on XLM-R+ fine-tuned on the annotated and augmented data as XLM-R+_AUG.

**Input Prompt**

The following tweets containing misinformation about vaccines belong to the same category **infertility**:

Example 1: The only reason why I'm not getting vaccinated is so I can donate my sperm for 1 mill
Example 2: Vaccine whistleblower says new GSK vaccine causes infertility in 97% of women http
...
Example 10: How many pregnant women have to lose their babies before the government acknowledges that vaccines should NEVER be given during pregnancy 😢 😢 😢 http
Example 11:

**Model Output**

My friend's cousin got the COVID-19 vaccine and now she can't get pregnant. Vaccines are dangerous and should be avoided. #vaccinesarepoison

Figure 1: Data augmentation process using GPT-3 with example input and generated output. Given a category and 10 example tweets, the prompt is constructed following the template above. The prompt is then fed into GPT-3 to obtain generated tweet to augment our dataset.

### 4.5 Evaluation Setting

In order to train and evaluate our model, we employ 5-fold cross validation. For hyperparameter search, we experiment on the combinations of batch size {8, 16, 32}, learning rate {1e-5, 2e-5, 3e-5, 4e-5, 5e-5}, and training epochs ranging from 1 to 10. The final models used in our large-scale study are trained on the entire annotated data with the best hyperparameters determined during the cross validation process. When applying text augmentation (described in §4.4), we combine the augmented data from each fold with the annotated data for training the final models.

We employ the macro F-1 score (averaged over the 5-folds) as the performance metric to evaluate our models as the classification accuracy is not a reliable measure for evaluating model performance

---

[3]The model is initialized with the weights of xlm-roberta-base from HuggingFace.

[4]We use GPT-3 text-davinci-003 with a temperature of 0.5 to generate tokens.

when the distribution of class labels in the dataset is highly skewed such as in our dataset (Table 1).

## 4.6 Experiment Results

Table 2 shows the performances for vaccine relevance (Q1) and COVID-19 vaccine relevance (Q2) models. Both models achieved high macro F-1 scores (i.e., a score of 97.8 and 92.9 for Q1 and Q2 respectively, when fine-tuning XLM-R+). Due to the high performance of the baseline XLM-R model and the relatively larger size of the dataset compared to the misinformation-related dataset (i.e., Q3 and Q4a to Q4h), the performance gain from domain-specific pre-training (i.e., XLM-R+) is smaller compared to the misinformation-related results presented in Table 3.

|    | XLM-R | XLM-R+ | Gain |
|----|-------|--------|------|
| Q1 | 97.0  | **97.8** | 0.8 |
| Q2 | 91.6  | **92.9** | 1.3 |

Table 2: Macro F-1 Score for vaccine relevance (Q1) and COVID-19 vaccine relevance (Q2) models. Best performance for each question is in **bold**.

As shown in Table 3, the models combining domain-specific pre-training and text augmentation strategy (i.e., XLM-R+_AUG) significantly outperform the baseline models (XLM-R), with the performance gain ranging from 2.7 to 15.9 percent points. In particular, we observe that the performance gain from text augmentation is more pronounced when the original annotated data is extremely imbalanced (i.e., Q4c and Q4h). This highlights the effectiveness of text augmentation using large language models to address challenges posed by imbalanced datasets.

|     | GPT-3 ZS | GPT-3 FS | XLM-R | XLM-R+ | XLM-R+_AUG | Gain |
|-----|----------|----------|-------|--------|------------|------|
| Q3  | 57.0 | 75.5 | 76.6 | 81.1 | **81.2** | +4.6 |
| Q4a | 45.5 | 53.1 | 77.9 | **80.7** | 80.6 | +2.8 |
| Q4b | 34.2 | 56.8 | 76.5 | 79.5 | **80.2** | +3.6 |
| Q4c | 49.4 | 58.1 | 83.8 | 84.9 | **87.7** | +3.9 |
| Q4d | 48.6 | 61.1 | 76.5 | 78.0 | **79.2** | +2.7 |
| Q4e | 55.5 | 78.2 | 80.1 | 84.0 | **85.0** | +4.9 |
| Q4f | 47.9 | 51.1 | 79.7 | 82.3 | **83.8** | +4.2 |
| Q4g | 43.1 | 67.8 | 77.7 | 80.6 | **81.8** | +4.2 |
| Q4h | 49.3 | 48.4 | 59.4 | 65.7 | **75.4** | +15.9 |

Table 3: Macro F-1 Score for GPT-3 Zero-Shot (GPT-3 ZS) and Few-Shot (GPT-3 FS), XLM-R, XLM-R+, and XLM-R+_AUG for Q3 and Q4a to Q4h. Best performance for each question is in **bold**.

We also evaluate GPT-3's zero-shot and few-shot classification performance on our dataset, averaged over the same 5-folds (§4.5). In the few-shot setting, we use K (=5) examples of tweets and label

pairs to construct an input prompt. To ensure that both positive and negative examples are evenly included in the input prompt, three (or two) positive examples and two (or three) negative examples are randomly chosen from the training set. The examples and details of the input prompts are described in Appendix A.2. Table 3 leftmost columns show the results for GPT-3 zero-shot (GPT-3 ZS) and few-shot (GPT-3 FS) classification performances. Our fine-tuned models (XLM-R+_AUG) outperform GPT-3 model performance on the dataset.

|     | Total w/ CI | Brazil | Indonesia | Nigeria |
|-----|-------------|--------|-----------|---------|
| Q1  | 97.85 ($\pm$ 0.44) | 97.26 | 97.00 | **98.34** |
| Q2  | 92.89 ($\pm$ 0.94) | **96.36** | 92.41 | 92.99 |
| Q3  | 81.25 ($\pm$ 1.88) | 72.25 | 76.99 | **81.05** |
| Q4a | 80.60 ($\pm$ 2.85) | **79.67** | 68.49 | 76.27 |
| Q4b | 80.16 ($\pm$ 1.19) | 77.54 | 75.44 | **82.19** |
| Q4c | 87.66 ($\pm$ 7.73) | **96.34** | 77.77 | 89.87 |
| Q4d | 79.15 ($\pm$ 3.15) | 80.80 | 74.95 | **81.84** |
| Q4e | 85.04 ($\pm$ 3.05) | 68.89 | 81.71 | **89.85** |
| Q4f | 83.84 ($\pm$ 2.47) | 69.87 | 77.85 | **83.93** |
| Q4g | 81.83 ($\pm$ 3.95) | 78.23 | 76.54 | **85.66** |
| Q4h | 75.36 ($\pm$ 15.15) | 78.79 | **81.38** | 77.90 |

Table 4: Macro F-1 Score of the best-performing model on the total dataset (with 95% Confidence Interval) and on each country-specific dataset. For Q1 and Q2, the best-performing model is XLM-R+, and for the rest is XLM-R+_AUG. For each Question, the country with the highest performance is in **bold**.

As we conduct an extensive analysis of tweets collected from three middle-income countries (§5), we also present *per-country* evaluations, focusing on the best-performing model for each question: XLM-R+ for Q1 and Q2, and XLM-R+_AUG for Q3 and Q4a to Q4h. As shown in Table 4, performance varies across countries. Notably, most models perform best on tweets from Nigeria, followed by Brazil and Indonesia. Further analysis suggests that this performance is closely linked to the language distribution of tweets within each country. Specifically, for Nigeria, 99.37% of the tweets were in English, followed by Brazil at 77.16%, and Indonesia at 45.59%. The enhanced performance, correlated with a higher English tweet ratio, can be attributed to the significant proportion of English in 1) the corpus used for XLM-R pre-training, 2) the tweets employed during the further pre-training of XLM-R+, and 3) the annotated tweets utilized in fine-tuning the base encoders.

| | Brazil | | | | Indonesia | | | | Nigeria | | | | Total |
|---|---|---|---|---|---|---|---|---|---|---|---|---|---|
| | 2020 | 2021 | 2022 | Country Total | 2020 | 2021 | 2022 | Country Total | 2020 | 2021 | 2022 | Country Total | |
| # of Tweets Collected | 722,494 (100) | 1,407,079 (100) | 3,395,071 (100) | 5,524,644 (100) | 1,574,567 (100) | 4,321,465 (100) | 1,563,274 (100) | 7,459,306 (100) | 2,827,268 (100) | 1,925,233 (100) | 1,072,780 (100) | 5,825,281 (100) | 18,809,231 (100) |
| # of Tweets Relevant to Vaccines | 211,081 (29.2) | 853,138 (60.6) | 2,733,095 (80.5) | 3,797,314 (68.7) | 674,589 (42.8) | 3,103,959 (71.8) | 793,079 (50.7) | 4,571,627 (61.3) | 375,667 (13.3) | 758,348 (39.4) | 148,089 (13.8) | 1,282,104 (22.0) | 9,651,045 (51.3) |
| # of Tweets Relevant to COVID-19 Vaccine | 201,536 (95.5) | 837,323 (98.1) | 2,580,232 (94.4) | 3,619,091 (95.3) | 646,823 (95.9) | 3,061,968 (98.6) | 746,436 (94.1) | 4,455,227 (97.5) | 333,055 (88.7) | 709,080 (93.5) | 119,881 (81.0) | 1,162,016 (90.6) | 9,236,334 (95.7) |
| # of Tweets Relevant to COVID-19 Vaccine Misinformation | 54,034 (26.8) | 210,237 (25.1) | 1,040,556 (40.3) | 1,304,827 (36.1) | 146,668 (22.7) | 502,708 (16.4) | 217,025 (29.1) | 866,401 (19.4) | 100,989 (30.3) | 169,146 (23.9) | 32,683 (27.3) | 302,818 (26.1) | 2,474,046 (26.8) |

Table 5: Summary of prediction results on the total collected tweets.

## 5 Analysis

For the final models, the best-performing model for each question, discovered during the cross-validation process (Table 2 and Table 3), is trained on the entire labeled data. Final models for Q3 and Q4a to Q4h are trained on the combination of labeled data and augmented data from all folds (as described in §4.4 and §4.5).

Subsequently, these final models are applied step-by-step from Q1 to Q4 to ∼19 million collected tweets. First, we predict whether a tweet is relevant to vaccines using our vaccine relevance classification model (Q1) and filter out tweets that are not relevant to vaccines. Then, we filtered out tweets that are predicted to mention non-COVID vaccines (Q2). Next, we use our COVID-19 vaccine misinformation classification model (Q3) to predict whether tweets contain misinformation. Through these sequential predictions, we extract COVID-19 vaccine misinformation from the collected tweets, and perform analysis on them.

### 5.1 Statistics of prediction results

Table 5 shows the summary of prediction results using our final models. In Nigeria, among the tweets collected, only 22.0% of tweets were predicted as relevant to vaccines. Upon further examination, in October 2020 in particular, we find that most of the tweets that contain the term "shots" are related to the EndSARS protest, which was a protest against the illegal violence by the Special Anti-Robbery Squad (SARS) that has spread throughout Nigeria. This shows that relying solely on the search query terms to identify vaccine-related tweets has limitations of including posts that are not actually relevant to the topic; highlighting the need for machine learning-based classification system.

Also, compared to the other two countries, Nigeria has a relatively lower proportion of *COVID-19 vaccine*-related tweets: 90.6% of vaccine-related tweets. Tweets related to *non*-COVID vaccines mention vaccines for HIV, Hepatitis B, HPV, Monkeypox, etc. This shows that Nigeria faces significant public health challenges beyond COVID-19, with diseases such as HIV and Hepatitis B posing risks to the public.

### 5.2 Percentage changes in new COVID-19 cases and COVID-19 vaccine misinformation

This section aims to examine the impact of daily percentage changes in new COVID-19 cases on the daily rate of COVID-19 vaccine misinformation in Brazil, Indonesia, and Nigeria. Our analysis entails a causal interpretation of the findings, contingent upon the fulfillment of standard assumptions inherent to a distributed lag model, including the exogeneity of (contemporaneous and lagged) daily rates of misinformation. The absence of the assumptions implies that the outcomes should be construed merely as indicative of an association. The model is as follows:

$$y_t = \beta_0 + \beta_1 x_t + \beta_2 x_{t-1} + ... + \beta_{15} x_{t-14} + u_t \quad (1)$$

The above model was separately ran for each country. Subscript $t$ represents date, and $y_t$ is COVID-19 vaccine misinformation rate, defined as $\frac{\#\text{tweets with vaccine misinformation}}{\#\text{tweets with any vaccine information}}$ in date $t$. $x_t$ is defined as $100 * \frac{\#\text{new cases in date t}}{\#\text{new cases in date t-1}}$, and $u_t$ is an error term. In addition to considering the immediate number of newly infected individuals, the historical number of cases may also contribute to the propagation of misinformation. Thus, we have incorporated lagged percentage changes up to the 15th day (or 14th lag) for each country. The number of new cases are drawn from WHO COVID-19 data (OWID) .

We exclude the early time periods in which the total number of COVID-19 cases in each country had not yet surpassed 1,000. This exclusion is necessary due to the lack of a well-defined percentage change in new cases during these periods with numerous instances of zero new cases. Furthermore, for this analysis, we truncate the data after the first quarter of 2022, as the increasing trends in misinformation rates suggest potential structural changes that undermine the stationarity assumption of the distributed lag model. However, it is worth noting

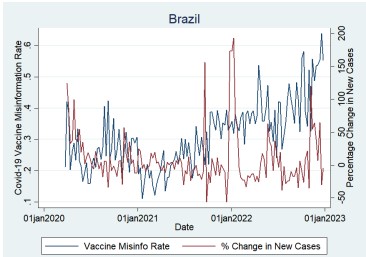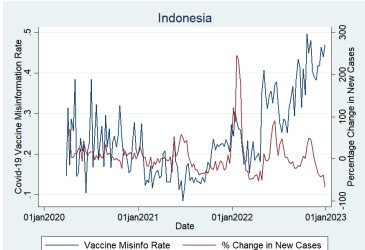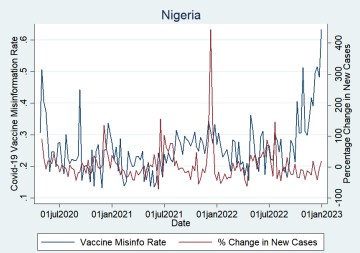

Figure 2: Weekly Time Series of Misinformation Rate and % Change in News Cases

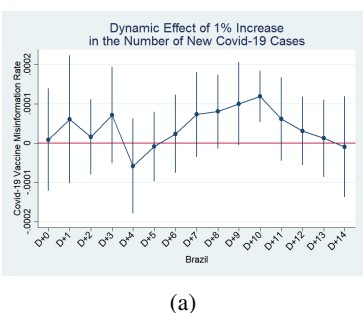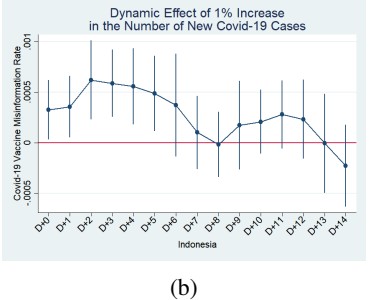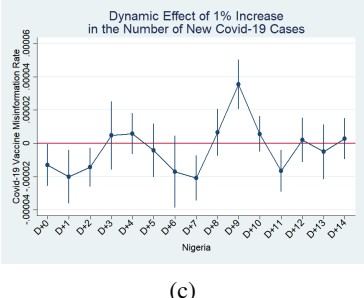

(a)            (b)            (c)

Figure 3: Dynamic Effect of 1% increase in the Number of New COVID-19 Cases

that the inclusion of these later periods does not significantly alter the qualitative results, and the Augmented Dickey-Fuller test confirmed that all six time series depicted in Figure 2 are stationary at a 0.01 significance level. Heteroskedasticity- and autocorrelation-consistent (HAC) standard errors are used for the inference.

Figure 3a illustrates the dynamic effect in Brazil with 95% confidence interval for each lag. The % changes in new cases take more than a week to affect misinformation rate in Brazil, and the effect subsides. For the early lags up to 7th lag, we cannot reject that the regression coefficients are jointly 0 (p=0.3896). However, the coefficients for 8th to 14th lags are jointly different from 0 (p-=0.0248). Specifically, the 10th lag has coefficient of approximately .0001, and it interprets as 1 percent increase in the number of new cases today increases misinformation rate by .0001 after 10 days. The coefficient is statistically significant but not effectively large in magnitude, considering that the mean and the standard deviation of misinformation rate in Brazil during the sample period are .2655 and .1101, respectively.

Figure 3b indicates that the % changes in new cases contributes to misinformation in Indonesia at a rate faster than Brazil. The magnitude of the dynamic effect increases until the 2nd lag and subsides to lose statistical significance from the 6th lag. The effect is much larger in magnitude, even when we only consider the early lags with statistical significance. The model estimates that 1% increase

in the number of new cases today increases misinformation rate by 0.0029 during the first 6 days. This is sizable compared to the mean and the standard deviation of misinformation rate of Indonesia, which are .1996 and .0936, respectively, where 32% increase in new cases amounting to almost 1 standard deviation increase in the misinformation rate. The coefficient increases again from the 8th lag and subsides after 11th lag, but we cannot reject that the coefficient for 8th to 14th lag are jointly 0 (p=.2951).

Figure 3c shows some coefficients with statistical significance, but these are small in magnitude and their signs are alternating. This inconsistent finding is likely attributed to the Nigerian government banning Twitter from 2021-2022 which restricted Nigerians from accessing Twitter. The government ban was due to political and economic factors and was lifted when Twitter agreed to legal and financial terms (BBC News, 2022).

|  | (1) BRA | (2) BRA | (3) IDN | (4) IDN | (5) NGA | (6) NGA |
|---|---|---|---|---|---|---|
| BRA |  |  | 0.189*** (0.045) |  | 0.261*** (0.047) |  |
| IDN | 0.261*** (0.084) |  |  |  |  | 0.200** (0.083) |
| NGA |  | 0.288*** (0.048) |  | 0.164** (0.067) |  |  |
| Constant | 0.214*** (0.016) | 0.194*** (0.013) | 0.149*** (0.014) | 0.159*** (0.016) | 0.174*** (0.012) | 0.203*** (0.016) |
| Observations | 703 | 674 | 703 | 674 | 674 | 674 |

Note: BRA, IND, and NGA indicate the misinformation rate of Brazil, Indonesia, and Nigeria, respectively. Column variables are dependent variables, and row variables are independent variables. Each column represents a different regression. HAC standard errors in parentheses. * 0.10 ** 0.05 *** 0.01.

Table 6: Coefficients from OLS Regressions Using COVID-19 Vaccine Misinformation Ratios

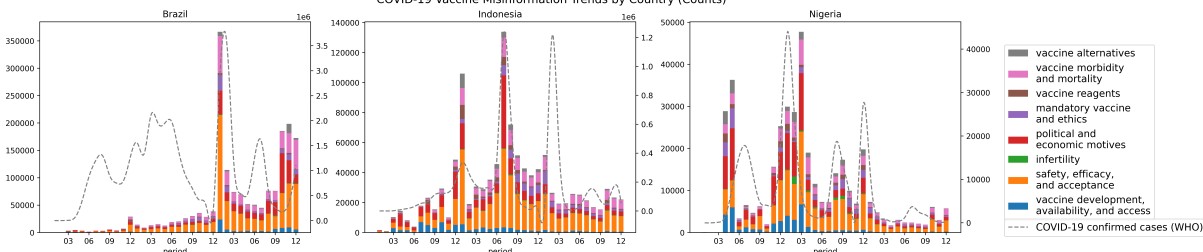

Figure 4: Trends of COVID-19 Vaccine Misinformation Themes by Country (Counts)

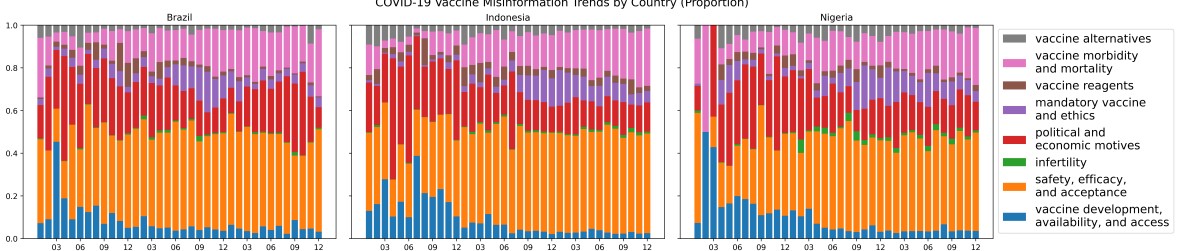

Figure 5: Trends of COVID-19 Vaccine Misinformation Themes by Country (Proportion)

### 5.3 Association of COVID-19 vaccine misinformation between countries

We hypothesize a positive association for vaccine misinformation rates across the three countries. We employed a simple linear regression model with HAC standard errors. The same data truncation is applied as in § 5.2, yielding different number of observations across specifications in Table 6. The truncation of the observations that are second quarter of 2022 or later should decrease the coefficients due to the co-rising trends in the period illustrated in Figure 2. Nonetheless, misinformation rates are positively associated at least at the .05 significance level in the truncated period.

### 5.4 Qualitative Analysis of Tweets

Figure 4 shows the number of tweets containing COVID-19 vaccine misinformation on a monthly basis, along with the monthly COVID-19 confirmed cases. The peak months for the number of tweets containing misinformation vary across countries and often coincide with different periods of rapid increase in confirmed COVID-19 cases, suggesting that the prevalence of misinformation may be correlated with the number of cases. This observation aligns with previous studies which have observed how misinformation thrives where people have little control over their environmental threats, which include the global pandemic (Nyilasy, 2019); or how misinformation tends to be more prevalent in the time of crisis when there is a lack of information needed to make emergency decisions

(Muhammed T and Mathew, 2022). It is also worth noting that a qualitative study of COVID-19 misinformation in Indonesia has observed that misinformation were prevalent during the early times of the pandemic where there was a lot of uncertainty about the disease and later on during the peak of the Delta-variant crisis that brought about a lot of cases and deaths (Sodikin, 2022). Understanding this phenomenon can potentially inform effective models for mitigating misinformation.

Figure 5 illustrates the monthly proportion of each misinformation theme. In all three countries, the proportion of the vaccine development, availability, and access theme rises and declines over time. This can be attributed to the prevalence of unverified information regarding vaccine development, trials, trial participants, and procurement *prior* to the introduction of vaccines (Jan - March 2021). Additionally, the safety, efficacy, and acceptance theme was the most dominant across all three countries. This may reflect concerns surrounding the rapid development of COVID-19 vaccines and their emergency use authorization.

In addition, we analyze tweets belonging to different themes and examine variations across countries. The analysis revealed that even within the same theme, there are cultural and political differences among countries that contribute to variations in tweets. This highlights the need for customized measures tailored to local circumstances to effectively address vaccine misinformation.

Tweets falling under political and economic motives often mentioned profit-driven pharmaceutical

companies developing vaccines despite the existence of potential COVID-19 treatments. Some tweets also suggested intentional virus spread by certain countries to enhance their negotiation power. In Indonesia, there was a significant number of tweets attributing the source of misinformation to domestic individuals or regions, indicating a fair amount of misinformation being produced and disseminated within the country. This has implications for public agencies, emphasizing the importance of not only addressing misinformation from external sources but also engaging with internal sources. On the other hand, in Nigeria, there were numerous tweets expressing distrust towards the government or political elites, suggesting that the government would misuse vaccines for personal or party-related purposes. This is contrary to the focus on misinformation regarding foreign countries or pharmaceutical companies in the other two countries. It serves as an important example demonstrating how different content within the misinformation theme can gain attention due to local political factors.

Furthermore, we discover distinct variations in the content of tweets belonging to the vaccine reagents/ingredients misinformation theme across countries. In Brazil and Nigeria, a considerable number of tweets discussed the topic of microchips embedded in vaccines to read people's minds or track individuals. In contrast, in Indonesia, there was a unique trend where tweets mentioned the use of vaccine ingredients that were religiously prohibited (not halal). Such misinformation reflects specific anxieties within the Indonesian context, where halal certification holds great importance for the Muslim population. It serves as a reminder that combating vaccine misinformation requires tailored and culturally sensitive approaches.

## 6   Conclusion

In this work, we present a multilingual dataset of tweets from three middle income countries annotated with their relevance to COVID-19 vaccines, the presence of misinformation, and themes of misinformation. Our method leveraging domain-specific pre-training and text augmentation demonstrate performance improvement for detecting misinformation and its themes. We also show applications of our COVID-19 vaccine misinformation system by conducting extensive analysis on a large corpus of tweets from 2020 to 2022. Distributed lag model indicates that % changes in the number of new COVID-19 cases increase misinformation rate over time for Brazil and Indonesia. We find positive association in misinformation rates between Brazil, Indonesia, and Nigeria. Additionally, our qualitative content analysis on tweets shows that vaccine misinformation can be contingent on local factors. We believe our dataset would allow public health communicators and researcher to discern the types of misinformation that are prevalent in each country and facilitate targeted communication strategies that are specifically tailored to local circumstances.

## Limitations

### Twitter ban in Nigeria

We acknowledge that there may be inaccuracies in analysis of data from Nigeria for the period from 2021 to 2022 due to the Twitter ban imposed by the Nigerian government (§5.2). Therefore, we plan to collect additional data on countries in the African region that are geographically, economically, and culturally similar to Nigeria (e.g., Ghana) in the future. This will enable comparative analysis of how communication on social media regarding misinformation differs between Nigeria, where there was external intervention in Twitter usage, and countries where such intervention did not occur.

### Suboptimal Model

Furthermore, while our models have outperformed competitive baselines including GPT-3 models and while we have qualitatively examined a large number of our model's predictions in our qualitative analysis (§ 5.4), we recognize that the model's performance may still be suboptimal. This is likely because our current methodology relies solely on the linguistic and semantic features of tweets. Therefore, in the future, we plan to benchmark methodologies that leverage external knowledge (Hu et al., 2021) or consider the external context surrounding tweets (Sheng et al., 2022), in addition to their inherent properties, on our data.

### Data Alterations for Analysis in §5.2 and §5.3

26 observations are interpolated to generate % change in new COVID-19 cases variable and to estimate HAC standard errors. These are 25 observations with zero new cases (2 from Brazil and 23 from Nigeria) and 1 missing observation (from Brazil). We used smoothed number of new cases to replace the zeros, and took an average of neigh-

boring number of new cases to replace the missing value. However, we believe that the influence of this alteration scheme should be marginal at best. Observations from Nigeria suffers more severely from the issue of zero new cases after the first quarter of 2022, but these were not used for the analysis.

**Causal Interpretation for Analysis §5.2 and §5.3**

The coefficients in §5.2 can be interpreted as (lagged) effects in a causal sense, given that the standard assumptions for the model are satisfied. We have made efforts to ensure that some of the assumptions are satisfied by truncating the analysis periods and running relevant tests. We do not intend the result in §5.3 to be interpreted as causal, because existence of confounding factors cannot be ruled out.

## Ethics Statement

The study protocol was reviewed by and determined to not pertain to human subjects research by the authors' Institutional Review Board. To comply with Twitter Content Redistribution policy, we will only release Tweet IDs/URLs and their annotations in our dataset. No other personal or identifiable information will be shared or published. All data were anonymized and de-identified prior to analysis to protect the privacy of the individuals who posted the tweets. The primary aim of this research is to contribute to the understanding of COVID-19 vaccine misinformation, and to inform strategies for public health communication.

## Acknowledgements

We thank anonymous reviewers for their helpful feedback on this work. This research was supported by Boston University's Center for Emerging Infectious Diseases Policy & Research (CEID), NSF CCF-2200052, and NSF grant 1838193. Any opinions, findings, conclusions, or recommendations expressed here are those of the authors and do not necessarily reflect the view of the sponsor.

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

# A Appendix

## A.1 Vaccine-Related Query Terms

We used the following search query: "vaccine OR vaccines OR vaccination OR vaccinate OR vaccinated OR jab OR jabs OR vax OR vaxxed OR anti-vax OR anti-vaxxer OR antivax OR antivaxxer OR inoculate OR vaxes OR vaxing OR vaxxes OR vaxxing OR vaxxie OR (shot NOT (party OR parties OR drinking OR drunk OR booze OR vodka OR tequila OR whisky OR alcohol OR wine OR beer OR hangover OR wasted OR hungover OR cocktail))OR (shots NOT (party OR parties OR drinking OR drunk OR booze OR vodka OR tequila OR whisky OR alcohol OR wine OR beer OR hangover OR wasted OR hungover OR cocktail))" We also included country-specific terms including "vacina," "injecao," and "Zé Gotinha" for Brazil; and "vaksin" and "suntik" for Indonesia per the Oxford Languages Word for Vax (Oxford Languages, 2021).

## A.2 Input Prompts for GPT-3 zero-shot and few-shot classification

Figure 6 and Figure 7 show the examples of the input prompts for detecting misinformation and themes of misinformation, respectively. For brevity, the few-shot examples and the corresponding labels are replaced with placeholders.

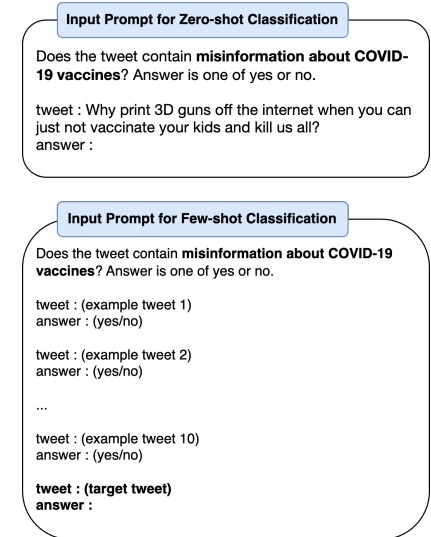

Figure 6: Input prompts for misinformation classification

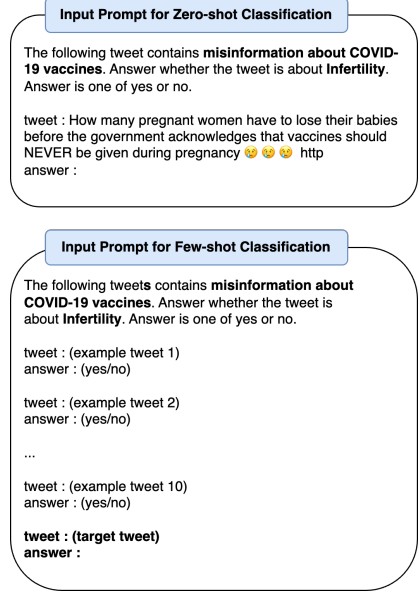

Figure 7: Input prompts for classifying themes of misinformation