# OpenReview forum: "COVID-19 Vaccine Misinformation in Middle Income Countries"
_EMNLP/2023/Conference — EMNLP 2023 Main_

### Official Review · Reviewer_Knzu · 2023-08-04

**Typos Grammar Style And Presentation Improvements:** The paper is well-written.
**Soundness:** 3

**Excitement:**

4: Strong: This paper deepens the understanding of some phenomenon or lowers the barriers to an existing research direction.

**Paper Topic And Main Contributions:**

This paper studies COVID-19 vaccine misinformation in three middle-income countries: Brazil, Indonesia, and Nigeria. The contributions are that the authors 1) curated a new dataset of tweets from these countries, labeled for COVID vaccine misinformation and its associated themes, 2) trained models to predict misinformation and its themes, 3) applied their trained models to all their collected tweets (~19 million) to analyze misinformation trends.

Dataset: first, they scrape all vaccine-related tweets from these countries (Jan 2020-Dec 2022). Then, they randomly sample a subset (N=5,500) to label. Each tweet is labeled for whether it is vaccine-related, whether it mentions non-COVID vaccine, whether it contains vaccine misinformation, and the theme of misinformation. Inter-coder reliability (ICR) is measured on 550 tweets that both annotators label; ICR is reasonably high.

Models: following their labeling scheme, they train a series of models to 1) predict if the tweet is vaccine-related, 2) predict if it is about the COVID vaccine, 3) predict if it contains misinformation, 4) predict the theme of misinformation (each theme is its own binary classifier). They employ two techniques to improve model performance: pre-training on domain-specific tweets and data augmentation via GPT-3. Their final model outperforms the base XLM-RoBERTA model.

Analyses: they apply their trained models to all their collected tweets (~19 million). First, they use a regression model to measure the impact of COVID-19 case rate on misinformation rate. They find small effects on some lags of COVID-19 case rate. Second, they find association between rates of misinformation of each country. Finally, they do a qualitative analysis and observe differences in language across countries, within the same theme (eg, political and economic motives).

**Questions For The Authors:**

A. In the modeling section, are there drawbacks to using a 4-step model pipeline, since errors could propagate? Did you consider merging some of the steps, eg, 1+2 -> directly predict if a tweet is about the COVID vaccine, or 3+4 -> directly predict if a COVID vaccine tweet is about a certain misinformation theme?
B. What is the motivation for predicting the impact of COVID cases on vaccine misinformation rates? Intuitively, I would think the relationship should be the other way around, ie, misinformation -> lower vaccine rates -> higher COVID cases?
C. I was a bit confused by lines 227-229: "To balance the data for misinformation, we identify more tweets that contain misinformation, resulting in the final annotated dataset of N=5,952." – what does this mean? How were more tweets identified?

**Reasons To Accept:**

- This is a thorough study, including a new dataset, predictive models, and analyses.
- The new dataset is richly labeled and interannotator agreement is high.
- The models achieve strong performance, with some nice techniques incorporated to outperform baselines.
- As the authors note, much of research on COVID-19 vaccine hesitancy focuses on the US, and it is important to study other countries.

**Reasons To Reject:**

- The new dataset is relatively small. While it contains 5,500 labeled tweets, only 3,011 are related to the COVID vaccine and 1,119 contain COVID vaccine misinformation. Furthermore, within that set, some themes are barely represented, eg, only 26 tweets for Q4C (infertility).
- Evaluation results in Section 4.5 are missing confidence intervals, which are especially important with small data.
- It would also be useful to provide evaluation per country, since the numbers here seem to be aggregated over all three countries. It is important to understand whether the model performs equally well across countries, especially if the model is then applied to analyze trends across countries.
- Unclear takeaways from the Analysis section. The effect sizes in 5.2 are very small and inconsistent across countries, and I'm not sure why we should expect COVID-19 rates to predict misinformation, instead of the other way around.

**Reproducibility:**

4: Could mostly reproduce the results, but there may be some variation because of sample variance or minor variations in their interpretation of the protocol or method.

**Reviewer Confidence:**

5: Positive that my evaluation is correct. I read the paper very carefully and I am very familiar with related work.

---

> ### Author Rebuttal · Authors · 2023-08-29
>
> Thank you for your positive recommendation and detailed review!
>
> ***Comment 1 - Relatively small dataset***
>
> We acknowledge that the dataset we have introduced might be relatively small. However, our multi-country dataset contains rich annotations, including relevance to vaccines and COVID-19 vaccines, presence of misinformation, and themes of misinformation, which requires greater annotation effort for each individual tweet. We believe these attributes of the dataset bestow it with significant value for the research of COVID-19 vaccine misinformation in middle-income countries.
>
> Additionally, you pointed out the scarcity of certain misinformation themes (e.g., infertility) in our dataset. This is mainly because our annotated dataset follows the theme distribution in the original population of tweets (i.e., tweets are randomly sampled and annotated from the total tweets we have scraped). The scarcity of observations in specific categories of misinformation themes is reflective of the data. To address this inherent limitation of the dataset (imbalance, some small classes), we introduce a text augmentation technique using a large language model, demonstrating strong performance improvements compared to the baseline models.
>
> ***Comment 2 & 3 - Confidence intervals & Evaluation per country***
>
> We appreciate your suggestion. We first provide the confidence intervals and evaluation per country only for the best models in our paper due to the space limit. We will include the full version of the table in the revision.
>
> |  | Confidence Interval (95%) | Brazil | Indonesia | Nigeria |
> |---|---|---|---|---|
> | Q1   | 97.85 $\pm$ 0.44 | 97.26 | 96.79 | 98.31 |
> | Q2   | 92.89 $\pm$ 0.94 | 94.38 | 91.24 | 92.58 |
> | Q3   | 81.25 $\pm$ 1.88 | 72.25 | 76.99 | 81.05 |
> | Q4a | 80.06 $\pm$ 2.85 | 79.67 | 68.49 | 76.27 |
> | Q4b | 80.16 $\pm$ 1.19 | 77.54 | 75.44 | 82.19 |
> | Q4c | 87.66 $\pm$ 7.73 | 96.34 | 77.77 | 89.87 |
> | Q4d | 79.15 $\pm$ 3.15 | 80.80 | 74.95 | 81.84 |
> | Q4e | 85.04 $\pm$ 3.05 | 75.84 | 79.08 | 79.98 |
> | Q4f  | 83.84 $\pm$ 2.47 | 69.87 | 77.85 | 83.93 |
> | Q4g | 81.83 $\pm$ 3.95 | 78.23 | 76.54 | 85.66 |
> | Q4h | 75.36 $\pm$ 15.15 | 78.79 | 76.34 | 73.49 |
>
> ***Comment 4 - Small and inconsistent effect sizes across countries***
>
> Concerning effect sizes, variations are observed across different countries. In the context of Indonesia, the (dynamic) effect size of 0.0029 is construed as signifying that a 1% escalation in the tally of new Covid cases brings about a 0.0029 increase in the misinformation rate (which ranges from 0 to 1) over an initial 6-day interval – an effect size deemed substantial. Illustrated in Figure 2 is a period at the onset of 2022 wherein the percentage surge in new case counts surpasses 200, translating to a dynamic effect size exceeding 0.58. In the case of Brazil, where the estimated effect size is very small, we can still learn that the Covid cases do not affect misinformation.
>
> ***Comment 5 - Drawbacks to using a 4-step pipeline since the errors could propagate***
>
> We thought this is an interesting idea, but followed the 4-step pipeline mainly due to the following reasons :
>
> Firstly, one of the main objectives of this paper is to introduce a new dataset and evaluate various models on this dataset. For this purpose, we developed and evaluated models following the 4-step coding scheme used in dataset construction. Additionally, while grouping steps as you suggested might reduce the propagation of errors, the problem of class imbalance could be even compounded. More importantly, following the 4-step pipeline enables a more comprehensive and detailed analysis on the ~19 million tweets as we could obtain predictions for all four steps.
>
> ***Comment 6 - Motivation for predicting the impact of (% change in) COVID cases on vaccine misinformation rates***
>
> Our approach of examining the influence of Covid cases on misinformation is based on the observed concurrent fluctuations in the (% change in) Covid cases and the rate of misinformation. The identical dynamic model which was employed to investigate the impact of Covid cases on misinformation was employed once more to investigate the relationship the other way around, yielding inconclusive results. This outcome was omitted due to space limitations, but we can add this information in the revision.
>
> ***Comment 7 - Clarification of how additional tweets were identified***
>
> As described in Section 3, the initial annotated dataset consisted of a total of 5,500 instances. However, considering the relatively low proportion of tweets containing misinformation, 452 tweets with misinformation were additionally incorporated into the dataset, resulting in a final dataset of 5,952 tweets. To include only tweets with misinformation, two coders identical to the previous round of annotation assessed the presence of misinformation in randomly sampled tweets from the collected dataset. Only those tweets containing misinformation were added to the annotated dataset (we balance the real/misinformation classes, not the themes). This approach is in line with previous works to balance real/fake classes in misinformation dataset (Sheng et al., 2022).
>
> Reference: Qiang Sheng, Juan Cao, Xueyao Zhang, Rundong Li, Danding Wang, and Yongchun Zhu. 2022. Zoom Out and Observe: News Environment Perception for Fake News Detection. In Proceedings of the 60th Annual Meeting of the Association for Computational Linguistics (Volume 1: Long Papers), pages 4543–4556, Dublin, Ireland. Association for Computational Linguistics.

---

### Official Review · Reviewer_XBsM · 2023-08-05

**Soundness:** 3

**Excitement:**

3: Ambivalent: It has merits (e.g., it reports state-of-the-art results, the idea is nice), but there are key weaknesses (e.g., it describes incremental work), and it can significantly benefit from another round of revision. However, I won't object to accepting it if my co-reviewers champion it.

**Paper Topic And Main Contributions:**

In this work the authors present a multi-language dataset for vaccine misinformation in middle income countries. The dataset is broken down into several smaller pieces that help with the global categorization of tweets to a more fine-grained level of different topics of vaccine misinformation. The evaluation is well designed and very methodical.

**Reasons To Accept:**

- Interesting multi-country and language dataset
- Methodical iterative use of tweets

**Reasons To Reject:**

- Inter annotator agreement is oddly calculated and seems quite low
- Extensive evaluation presented with data augmentation seems to be beyond the point of the individual dataset
- The paper is more than just a dataset release paper, making the narrative confusing

**Reproducibility:**

2: Would be hard pressed to reproduce the results. The contribution depends on data that are simply not available outside the author's institution or consortium; not enough details are provided.

**Reviewer Confidence:**

4: Quite sure. I tried to check the important points carefully. It's unlikely, though conceivable, that I missed something that should affect my ratings.

---

> ### Author Rebuttal · Authors · 2023-08-29
>
> Thank you for your feedback!
>
> ***Comment 1 - “Inter annotator agreement is oddly calculated and seems quite low”***
>
> Both Cohen’s Kappa and Gwet’s AC1 could be employed to evaluate the degree of agreement among raters. However, Gwet’s AC1 offers advantages over Cohen’s Kappa when dealing with binary and imbalanced data. Given that the majority of our coding scheme consists of questions utilizing binary response options (yes/no) and class imbalance is observed, we adopted Gwet’s AC1 over Cohen’s Kappa to ensure robust calculation of inter-annotator agreement.
>
> ***Comment 2 and 3 - The goal and narrative of the paper***
>
> The purpose of this study is not only to introduce a new dataset, but also to establish the initial baselines for the research that utilizes this dataset and to demonstrate practical applications of this dataset. To this end, several baseline methods as well as methods employing domain-specific pre-training and text augmentation were evaluated on the dataset. The best-performing model was then applied to a large set of unlabeled tweets for extensive analysis.
>
> While we have outlined the purpose and contributions of this study in the abstract and in the final paragraph of the introduction, we will also make this clearer in the revision so as to avoid confusion.

---

### Official Review · Reviewer_cr9A · 2023-08-13

**Soundness:** 4

**Excitement:**

4: Strong: This paper deepens the understanding of some phenomenon or lowers the barriers to an existing research direction.

**Justification For Ethical Concerns:**

The authors claims that their work is approved by their IRB.

**Missing References:**

[1] Karishma Sharma, Yizhou Zhang, and Yan Liu. Covid-19 vaccine misinformation campaigns and social media narratives. In Proceedings of the International AAAI Conference on Web and Social Media, pages 920–931.

[2] Yizhou Zhang, Defu Cao, and Yan Liu. Counterfactual neural temporal point process for estimating causal influence of misinformation on social media. In Advances in Neural Information Processing Systems 2022.

**Paper Topic And Main Contributions:**

This paper presents a multilingual COVID-19 vaccine misinformation in middle income countries. It mainly studies the public Twitter posts from three contries. The authors provide detailed annotations to a subset of the posts, such as domain-expert misinformation labelling and identified themes. They applied this annotated subset to train a misinformation detection model and then analyze the rest unlabelled data with the model. Their analysis reveals some findings about misinformation of COVID-19 vaccines and other factors.

**Questions For The Authors:**

Did you consider the probable confounders in Sec 5.2?

**Reasons To Accept:**

1. This research is really meaningful. Previous research mainly focus on the vaccine misinformation in specific languages in specific countries or cultures. This research first propose to focus on middle income countries.

2. The experiments are extensive (although with some issues that I will point out in the reasons to reject). The decriptions are detailed and provide sufficient information to understand

3. The annotations are detailed and will be very helpful for future works on this direction.

**Reasons To Reject:**

My main concern is in the experiment:

The causal effect of COVID-19 cases on misinformation. The authors claims that their analysis shows that increased COVID-19 cases are  positively affecting the vaccine misinformation ratio. This is problematic. My understanding is that their linear regression model does reveals a correlation. However, correlation may not be the causality. In causality area, the experiment that the authors did is treatment effect estimation. It is usually done after we have exclude the hypothesis of correlation, i.e. there might be some confounders that jointly influence the x and y. To make the claim in the paper hold, we need to first exclude the hypothesis of correlation. For example, we can do an IPTW to rebalance the data to make them ``seem" to be drawn from random controlled trial to reduce the influence of counfounders. Or the authors can rephrase their expression as "We found a positive correlation between ......". I provided two papers in the missing reference that may be helpful. [1] did some analysis to vaccine hesistancy and misinformation of different states in the U.S., but they did not claims misinformation ``affect" accine hesistancy because polictic trend could be a counfounder. Their following work [2] studies on some causal effect on social media.


**Reproducibility:**

4: Could mostly reproduce the results, but there may be some variation because of sample variance or minor variations in their interpretation of the protocol or method.

**Reviewer Confidence:**

3: Pretty sure, but there's a chance I missed something. Although I have a good feel for this area in general, I did not carefully check the paper's details, e.g., the math, experimental design, or novelty.

---

> ### Author Rebuttal · Authors · 2023-08-29
>
> Thank you for your positive recommendation and feedback!
>
> ***Comment 1- The causal effect of (% change in) COVID-19 cases on misinformation confounders***
>
> It is recognized that the application of linear regression may not elucidate causal relationships unless the requisite assumptions underpinning the model are met. Consequently, we will rephrase it in the following way: "A positive correlation has been identified between ..., which may potentially be construed as indicative of a causal link."
>
> A recommended course of action, as suggested by the reviewer, entails the inclusion of control variables to effectively address the issue of confounding. However, identifying suitable control variables has proven challenging, with the exception of policy stringency, as any variable that does not vary over time is absorbed into the model's constant term. Furthermore, even in instances where a variable exhibits temporal variance, its dynamic relationship presents formidable complexities. For instance, the impact of policy stringency on Covid cases might not manifest instantaneously but could materialize over the course of several days or weeks. Consequently, the decision was made to adopt a parsimonious model, while still taking into account potential heterogeneity and autocorrelation using Newey-West standard errors.

---

### Meta-Review · Area_Chair_u9Cy · 2023-09-18

**Recommendation:** 5

**Metareview:**

This paper describes the creation of a novel COVID-19 multilingual dataset that covers multiple middle-income countries. The dataset creation is sound and with high potential for impact given the lack of datasets of this nature.

The paper is primarily a dataset paper with benchmark experiments using off-the-shelf methods. The novel contributions on the experimental side are more limited, but the insights derived from these experiments as well as the benchmark performance scores are useful for future work, in addition to the key contribution of a dataset.

I would suggest the authors to take into account reviewer comments in a further revision by toning down the correlation claims and by adding a discussion on per-country performance scores.

---

### Decision · Program_Chairs · 2023-10-07

**Decision:**

Accept-Main

**Comment:**

This paper describes the creation of a novel COVID-19 multilingual dataset that covers multiple middle-income countries. The dataset creation is sound and with high potential for impact given the lack of datasets of this nature.

The paper is primarily a dataset paper with benchmark experiments using off-the-shelf methods. The novel contributions on the experimental side are more limited, but the insights derived from these experiments as well as the benchmark performance scores are useful for future work, in addition to the key contribution of a dataset.

I would suggest the authors to take into account reviewer comments in a further revision by toning down the correlation claims and by adding a discussion on per-country performance scores.